# Air-Launch Experiment Using Suspended Rail Launcher for Rockoon

**Tadayoshi Shoyama** [1,*] **, Ayana Banno** [2] **, Yousuke Furuta** [3] **, Noboru Kurata** [4] **, Daisuke Ode** [5] **, Yutaka Wada** [1] **and Takafumi Matsui** [1]

[1]   Planetary Exploration Research Center, Chiba Institute of Technology, Narashino 275-0016, Chiba, Japan; yutaka.wada@p.chibakoudai.jp (Y.W.); matsui.takafumi@p.chibakoudai.jp (T.M.)
[2]   Graduate School of Engineering, Chiba Institute of Technology, Narashino 275-0016, Chiba, Japan; s1671033kw@s.chibakoudai.jp
[3]   AXS Corporation, Shimonoseki 750-0323, Yamaguchi, Japan; yousuke.furuta@axs-jp.com
[4]   Sowa Engineering Corporation, Fujisawa 251-0016, Kanagawa, Japan; noboru-kurata@sowamd.co.jp
[5]   SPACE COTAN Co., Ltd., Hiroo 089-2113, Hokkaido, Japan; ode-daisuke@spacecotan.com
[*]   Correspondence: shoyama.tadayoshi@p.chibakoudai.jp

**Abstract:** The method of air-launching a rocket using a launcher suspended from a balloon, referred to as a rockoon, can improve the flight performance of small rockets. However, there have been safety issues and flight trajectory errors due to uncertainty with respect to the launch direction. Air-launch experiments were performed to demonstrate a rail launcher equipped with a control moment gyroscope to actively control the azimuth angle. As a preliminary study, it was suspended via a crane instead of a balloon. The rockets successfully flew along the target azimuth line and impacted the predicted safe area. The elevation angle of the launcher rail exhibited a fluctuation composed of two frequency components. A double-pendulum model with a rigid rod suspended by a wire was proposed to predict this behavior. Significant design parameters and error sources were investigated using this model, revealing the constraining effect of a large mass above the wire and elevation angle fluctuation, which caused trajectory errors due to the friction force on the rail guide and thrust misalignment. Finally, tradeoffs in designing the rail length were found between the launcher clear velocity and elevation fluctuations.

**Keywords:** rockoon; balloon; launcher; attitude dynamics; elevation angle; double pendulum

## 1. Introduction

There is an increasing demand for low-cost and small launch vehicles for scientific suborbital missions and dedicated orbital launches of small satellites. A large rocket is suitable for launching many satellites into a single orbital plane to form a constellation. However, small rockets are required for the dedicated launching of a small number of payloads into various orbits, including scientific missions. The world's smallest orbital launch vehicle is the Japanese SS-520-5, with a mass of 2.6 t [1,2]. It comprises three solid-propellant stages. However, the payload ratios of small rockets are lower than those of large rockets. The ratios can be significantly increased by performing air launches, where the rocket is launched into the air from an airplane or balloon [3,4]. Airplane-type air launches have been used throughout the history of rocket launches. The idea originated from an air-launched ballistic missile, Bold Orion and High Virgo, developed in the 1950s [5]. The Pegasus rocket is the first orbital rocket that was air-launched from an airplane and many commercial flights have been performed. Recently, Virgin Orbit performed the first successful air-launch of a liquid-propellant orbital rocket. Thus, theories and technologies for airplane-type air launches have been established at a practical level.

A method of launching a rocket using a launcher suspended from a balloon is referred to as rockoon. It presents the advantage of launching from a high altitude exceeding 20 km,

which is not possible by using civil aircraft. Because of the limitation of the balloon's volume, rockoon only applies to very small satellites and suborbital missions [6]. The problems associated with rockoon have been addressed in previous studies. The first experiment of rockoon was performed by a group under Van Allen in the 1950s with the aim of performing suborbital science missions for studying high-altitude atmospheres. The launcher was not equipped with a device to control the azimuth angle during launch. Hence, the balloon had to be launched from a vast desert or a ship that was isolated to the maximum extent from inhabited islands. Sigma rocket [7] was a Japanese rockoon that reached an altitude of 105 km in 1961. However, the program ended because of cost issues. In the USA, JP Aerospace [8] and California Maritime Academy [9] used tube-shaped launchers that were lifted by multiple balloons. The former rockoon performed successful launches while the latter exhibited problems related to limited helium supply. In Australia, HARP performed some rockoon launches [10]; however, the uncertain winds in Woomera posed problems in releasing a balloon and they concluded that the location was unsuitable for rockoon launches. Single- and two-stage rockets were successfully launched from a rail launcher lifted by balloons (developed at the University of Washington [11]), allowing the entire launcher system to be recovered, undamaged, to the ground in the single-stage mission. Although rockoon technologies have been gradually improved, major problems still exist in the attitude control of the launcher in the air and the high costs of solid propellant rockets and helium gas [12].

With respect to attitude control, the aforementioned rockoon systems used a suspended launcher without an attitude controller. In the HIMES project in Japan [13], the gondola suspended by a balloon was equipped with a reaction control system that used gas jet thrusters to control the azimuth angle. The rocket was ignited during free fall, soon after release. Because it is difficult to accurately predict the attitude of a falling rocket at ignition, the application of a rail launcher is preferable. However, the issue of the attitude disturbance of a suspended launcher still exists, especially in the azimuth angle. Wind gusts have small effects on the attitude disturbances because the gusts in the stratosphere are weak and the relative speed of a balloon to the air is small; however, the azimuth angle is easily affected by a small perturbation and pendulum motion of the balloon because it does not have an equilibrium angle. Nevertheless, the azimuth angle has never been controlled in previous rockoon projects. Attitude control technologies of gondolas on balloons have been researched in the balloon-borne telescope system [14,15]. A suspended telescope was oriented to the target direction with an accuracy of one arcminute using a control moment gyroscope (CMG) as the attitude control device. There exists a coupling of pendulum and azimuth dynamics through the azimuth control loop and a complete dynamic model of balloon-borne systems was proposed [16].

There are two differences between a balloon-borne telescope and a rockoon launcher. First, the center of gravity (CG) of the rockoon launcher moves as the rocket slides on the launcher rail and the mass suspended on the balloon decreases after the rocket leaves the launcher, causing dynamic behavior with respect to the elevation angle. Second, a friction force at the sliding point between the rocket and launcher leads to fluctuations in the attitude of the launcher. Even in ground launches, the nose-down movement of a rocket leaving from the end of an inclined rail launcher significantly affects the trajectory parameters, such as apogee altitude, impact point downrange, and maximum dynamic pressure. This becomes more significant for a suspended rockoon launcher because the attitude of the rail is not constrained to the ground. Thus, the dynamics and attitude control of a suspended launcher at launch is a major problem associated with rockoon; however, these problems have never been investigated.

In our rockoon project, we aim to launch a hybrid rocket from a stratospheric balloon for suborbital science missions and future satellite launch missions. A single-stage rocket with a mass of 190 kg will be ground-launched in 2022. It has been converted to an air-launch configuration to be launched from a balloon. They are finally extended to a three-stage orbital rocket with a mass of 771 kg. As reported in our previous study [17], a

hybrid rocket using $N_2O$ propellant is suitable for a high-altitude rockoon, because $N_2O$ can be stored in liquid state at the temperature of the atmosphere, preventing the heat conduction through the tank wall. The tank pressure decreases as the propellant is cooled by the atmosphere. The lower limit is 0.7 MPa, that is, the saturation pressure at the air temperature at 20 km of altitude. It is too low for self-pressurized feeding; therefore, a small battery-powered turbopump is also under development at CIT. The recent progress in low-cost hybrid rocket propulsion will accelerate the long-lasting development of rockoon systems because the non-combustible nature of hybrid rockets provides a significant advantage in terms of safety, with respect to the balloon or launcher accidentally falling. However, the aforementioned issues related to the attitude of the launcher have never been addressed.

This study has two goals. The first is to demonstrate the concept of a CMG-controlled rail launcher using air-launch experiments. The attitude was controlled by an operator and the experiment focused on the feasibility of applying the attitude controller to the suspended launcher system. The second goal is to clarify the fluctuation behaviors of the launcher's attitude at the launcher-clear, especially the elevation angle, which have significant effects on the flight trajectories. A simple dynamical model is proposed that reproduces the fluctuations observed before and after launch. Significant design parameters and error sources were discussed using the model.

## 2. Experiment

A rail launcher was used to maintain the attitude of the rocket during launch. It was equipped with a CMG device to control the azimuth angle. The launcher was lifted via a crane instead of a balloon because it is easy to control the launch point and altitude. Furthermore, its dynamic behavior can be observed closely from the ground. The drawbacks of using a crane is that the vertical motion is restricted, although an oscillation along the vertical axis is an issue in a high-altitude balloon [18]. A genuine balloon experiment is necessary in the future study; however, the goals of this study can be achieved even in the crane experiment.

The attitude controller of the launcher is necessary to maintain the target direction of the rail against disturbances caused by wind and balloon motion. With respect to elevation angle, there exists an equilibrium point at which the CG of the launcher, combined with the rocket, is on the extension line of the suspending wire. However, the azimuth angle should be actively controlled until the rocket clears the launcher because there is no equilibrium point in the azimuthal direction. Even if the launcher attitude is accurately controlled before ignition, as the rocket moves on the rail after ignition, the launcher attitude changes due to the frictional force at the sliding point and movement of the position of the CG. It is difficult to actively control the launcher attitude during this period because it is as short as 0.5 s. Such behavior should be predicted and considered when the target angles are determined before launch.

A small rail launcher, shown in Figure 1, was constructed to study the dynamic behavior during an actual launch. It was equipped with a CMG to control the azimuth angle of the launcher. The attitude of the launcher was recorded using an accelerometer mounted on top of the attitude controller. Figure 2 shows the details of the attitude controller structure. It comprises a pair of momentum wheels driven at the same angular velocity $\omega$ by electric motors located on both sides of the horizontal arms. The rotating axis is tilted by $\theta$ around the horizontal arm. A hobby servo motor and radio controller were used to enable remote-controlled operation. They were powered by a lithium polymer battery. Because the wheels were rotating in opposite directions to each other, the tilting direction was opposite to produce an azimuthal torque, as shown in Figure 2. If the rotational direction of the wheels is the same, the tilting angle should be the same. However, this type of motion generates a repulsive torque with respect to the horizontal arm, thereby leading to an unstable oscillation in the elevation angle. As the rotational speed of the wheels increases with angular momentum, the reaction torque obtained by tilting the rotation axes increases.

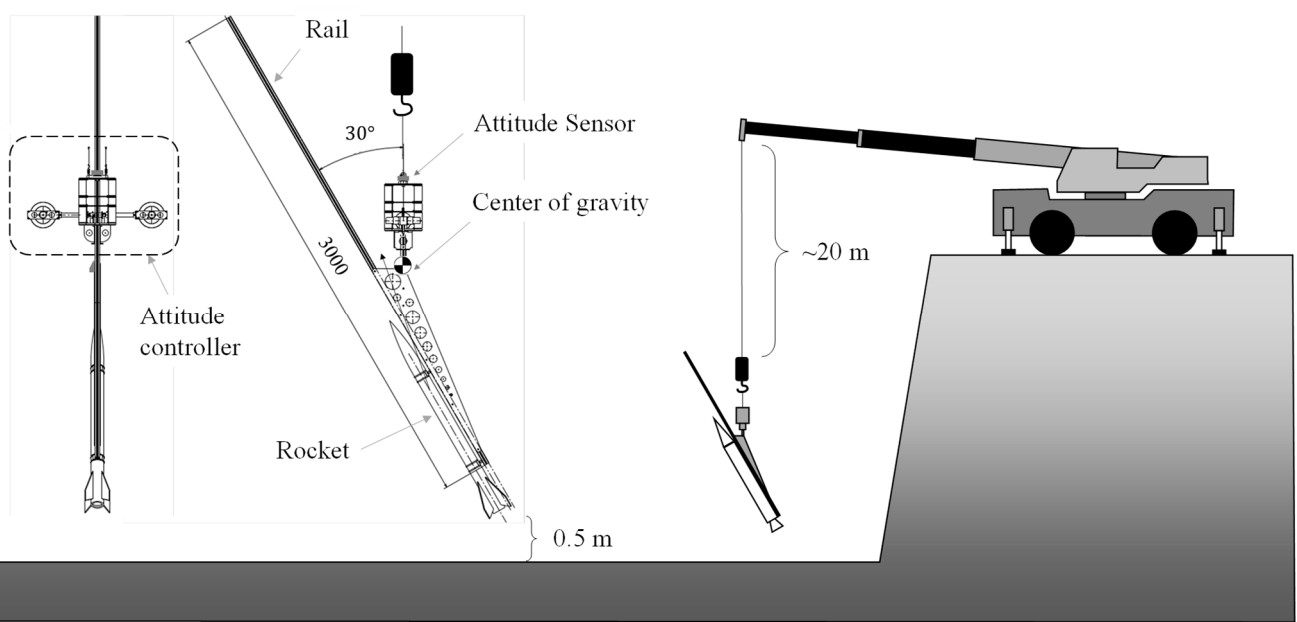

**Figure 1.** Rail launcher suspended via a crane.

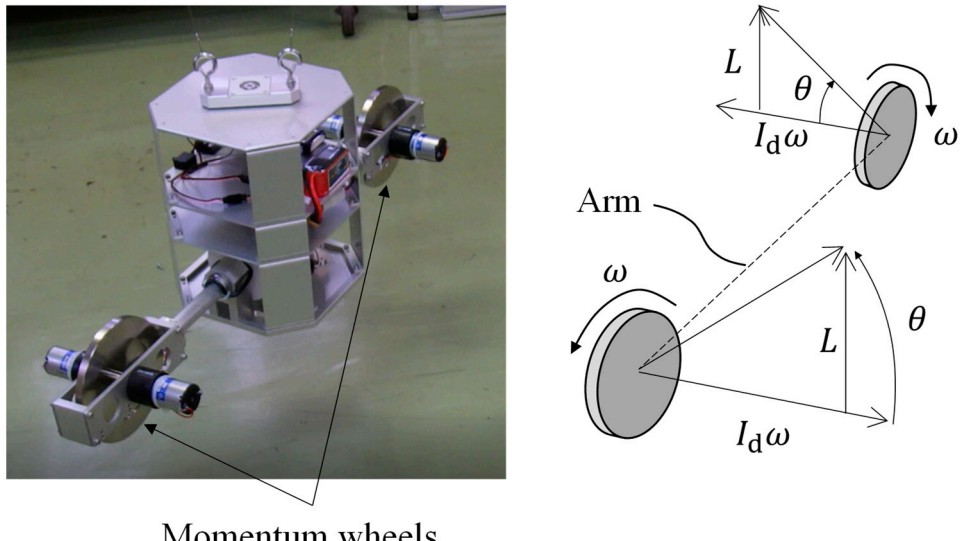

**Figure 2.** Structure of the attitude controller.

The dimensions and rotational speed of CMG were determined as follows. To calculate the generated torque, the initial state depicted in Figure 2 is considered. Suppose that both wheel axes are horizontal and rotated in opposite directions at the same angular velocity $\omega$. The angular momentum of the wheel is $I_d\omega$, where $I_d$ denotes the moment of inertia of the rotor assembly. When the rotor axes are tilted by $\theta$, the angular momentum of each wheel changes by $L = I_d\omega \sin\theta$ in the vertical upward direction and by $I_d\omega \cos\theta$ in the horizontal direction. Thus, the total momentum in the vertical direction increases by $2L$ while the change in the horizontal direction cancels out with each wheel. As a reaction, the launcher obtains the vertical downward angular momentum as follows:

$$2L = 2I_d\omega \sin\theta \tag{1}$$

When the launcher has a moment of inertia $I_L$ about the vertical axis, it starts clockwise rotation, when viewed from the above, at an angular velocity of

$$\Omega = 2\frac{I_d}{I_L}\omega\sin\theta \tag{2}$$

If it rotates before tilting, the rotation speed changes by $\Omega$. When the launcher is oriented toward the target azimuth, the tilted angle should revert to the original state to stop the rotation. Supplemental Video S1 shows this behavior. When it receives a unidirectional disturbance torque from the wind, it is necessary to continuously produce a canceling torque $M$. When the tilting angle rate is $\dot{\theta}$, the azimuthal torque generated by the CMG is obtained by differentiating Equation (1), as follows:

$$M = 2\dot{L} = 2I_d\omega\cos\theta\,\dot{\theta} \tag{3}$$

The tilting angle rate $\dot{\theta}$ must be applied continuously to cancel the steady azimuthal disturbance torque. When the tilting angle reaches $\theta = 90°$, the azimuthal torque $M$ vanishes and the rotational speed of the launcher cannot be changed further; this is CMG saturation. In this study, the specifications were determined as $I_L = 1.223$ kg·m$^2$, $I_d = 3.286 \times 10^{-3}$ kg·m$^2$, and $\omega = 135$ rad/s. Then, the maximum rotational speed at saturation is $\Omega = 0.725$ rad/s $= 41.6°$/s, which seems sufficient for attitude control. To increase the torque $M$, the wheel should be rotated as quickly as possible. The generation of the azimuthal torque is possible even when $\theta = 90°$ by changing the wheel speed $\omega$ at a rate $\dot{\omega}$. However, the generated torque $I_d\dot{\omega}$ was quite low and the duration required to maintain the torque was short because $I_d \ll I_L$ holds.

The launcher was not equipped with an active control function for the elevation angle. The launcher rail was fixed to an azimuthal attitude controller. The elevation angle fluctuates around the equilibrium point. After the launch, the CG changes with the attitude in equilibrium and oscillates about the equilibrium, similar to a pendulum. This behavior is discussed in detail later in this paper.

## 3. Experimental Results

### 3.1. Attitude Controller Unit Test

An operational test of the CMG attitude controller was performed without a launcher rail. Supplemental Video S1 shows that attitude control was accurately performed, thereby realizing controlled azimuthal rotation. To estimate the moment of inertia of the motor shaft that drives each momentum wheel, the same experiment was performed without the momentum wheels. As a result, no effective rotational torque is generated. Hence, most of the angular momentum of the CMG is generated by the momentum wheels and the contributions of the motor and shaft are negligible.

### 3.2. Air-Launch Trajectory

The attitude controller is fixed to the rail launcher. The launch experiments were performed at a quarry at Ube Kyouritsu Sangyou, Inc., in Yamaguchi, Japan. There is an ellipse-shaped depression, which is 400 m long, 200 m wide, and surrounded by hard bedrock. As depicted in Figure 1, the launcher is suspended 0.5 m above the ground using a crane, which is placed 13 m above the launch point. Considering the possibility of the malfunction of the attitude controller, in which the rocket could fly in an arbitrary direction, this type of environment was selected as a safe launch point. The length of the crane wire was $l = 20$ m; the frequency of the simple pendulum was $2\pi\sqrt{l/g} = 9$ s, which is much longer than the time scale of the launching motion.

The launcher assembly, which comprised a CMG attitude controller and rail launcher of 3 m length, was suspended by a wire of length $l_1 = 0.357$ m on a crane hook having a mass of 60 kg. The elevation angle was fixed at 60°. The model rocket was a Spitfire PML 30,385 (diameter: 76 mm, height: 1288 mm, and mass: 1.16 kg). A three-axis angular

velocity sensor was attached to the top of the attitude controller. The rocket was installed on a suspended launcher after preparation. After an evacuation procedure, the CMG started to control the attitude of the launcher. As soon as the launcher was oriented toward the target azimuth, the rocket was ignited remotely. The target azimuth was set along the major axis of the depression. Three launches were performed with solid motors with a nominal total impulse of 137 Ns. For the first and second launches, the propellant mass was reduced by half for safety purposes. Supplemental Video S2 shows the second air-launch experiment. The locations of the launch and impact points are shown in Figure 3. The flight distances ranged from 45 to 147 m and the azimuth angles of the impact points were included in the range of 8°; i.e., 1.5° southward error and 6.5° northward error. The impact points are located northward of the target azimuth line because of the weak southern wind.

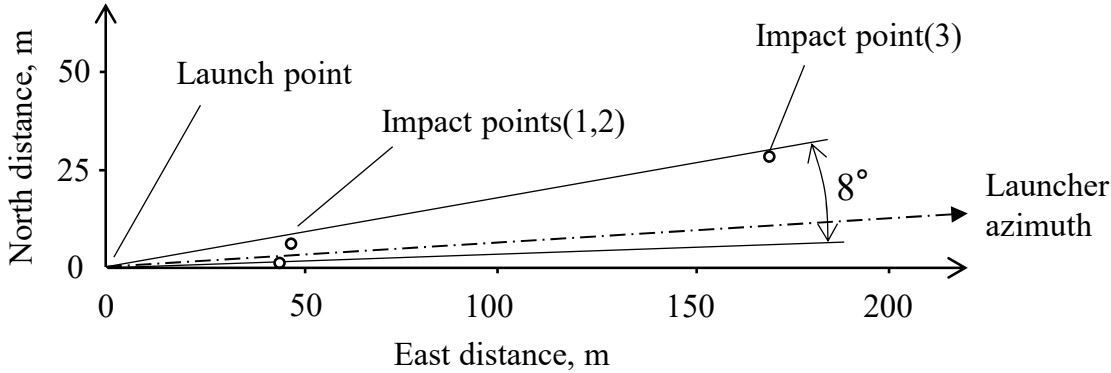

**Figure 3.** Location of the launch point and impact points.

### 3.3. Launcher Attitude

Figure 4 shows the attitude histories of the launcher measured by the attitude sensor. The measured angular velocities were integrated to obtain the attitude angles. Hereafter, the ignition time is referred as X. In the first launch, attitude control was started at X − 45 s and the azimuth maneuver ended at X − 17 s. In the second launch, the attitude was maintained from X − 30 s. The azimuth angle was controlled by an operator and maintained in the $85 \pm 8°$ range in the first launch. The azimuth control accuracy was improved to $84 \pm 4°$ in the second launch, owning to the improvement of the operator's skill. The azimuthal error in the first launch was the largest in the other launches. For a full-scale rockoon system, such an error will have a significant effect on enlarging the impact area or the orbit insertion error; therefore, it should be reduced by installing an autonomous controller in the future. After the launch, the attitude control was stopped at X + 5 s. No azimuthal perturbation was observed at ignition in both launches.

The elevation angle oscillated with a peak-to-peak amplitude of 2° about the equilibrium until ignition. After ignition, the equilibrium elevation angle decreased by 4.4° and a non-sinusoidal oscillation with a peak-to-peak amplitude of 10° was excited with respect to the new equilibrium elevation angle. This type of behavior is characteristic of suspended launchers. The decrease in the equilibrium angle was in good agreement with an angle of 5.4°, which was predicted based on the movement of the CG of the launcher assembly after the removal of the rocket. The largest elevation was observed at X + 0.8 s after the ignition; this was due to an upward moment owing to the friction force on the rail and the thrust misalignment force. This effect increased the actual elevation angle by 0.8°. Approximately the same increase in the elevation angle was observed in the other launches.

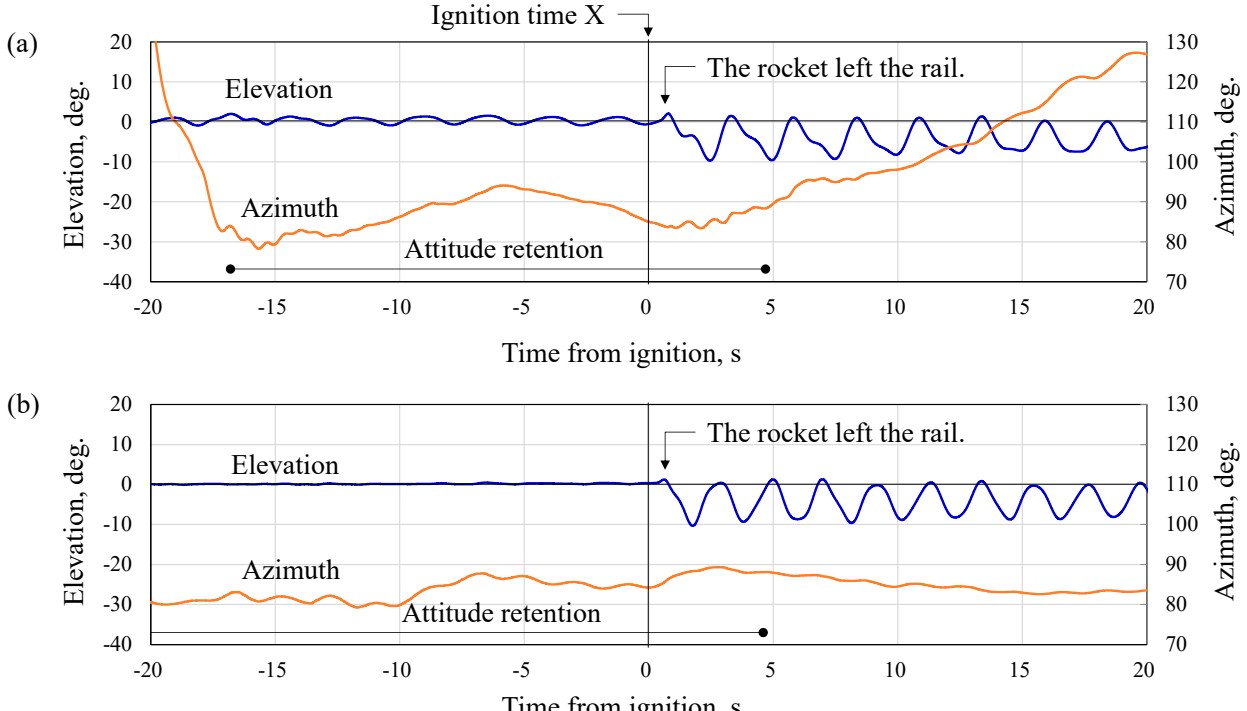

**Figure 4.** Launcher attitude with respect to time. (**a**) First launch (**b**) Second launch.

## 4. Discussion

The experimental results exhibited an increase in the actual elevation angle when the rocket reached the end of the launcher rail. This behavior changes the actual elevation angle and significantly affects the trajectory accuracy and range safety; therefore, detailed discussions on these pendulum dynamics are necessary to predict the change in elevation angle. Although a complete dynamic model of balloon-borne systems was proposed by Kassarian [16], relative motions and separations of an internal object cannot be considered. In this section, a pendulum model is proposed for a suspended launcher with a launch object. The dynamics after the rocket has left the launcher seems to be of little importance because it does not affect the launch trajectory. However, the detailed discussion is necessary to validate the model and to show the reader how to identify the model parameters.

### 4.1. Double Pendulum Model

To obtain a dynamical model of the suspended launcher, we focused on the non-sinusoidal oscillation of the elevation angle, which was observed after ignition. Because these behaviors were similar in both launches in Figure 4, only the first launch is considered in the following discussions. Figure 5 shows the results of the fast Fourier transformation (FFT) of the elevation angle for the two 14-second time series before ignition (X − 15.3 s–X − 1.3 s) and after ignition (X + 0.5 s–X + 14.5 s). In both cases, two components of frequency $f_1$ and $f_2$ were observed. The frequency of the lower component, $f_1$, increased after the ignition because the CG moved upward owing to the absence of the rocket, reducing the length of the pendulum. However, the frequency of the other component, $f_2$, decreased.

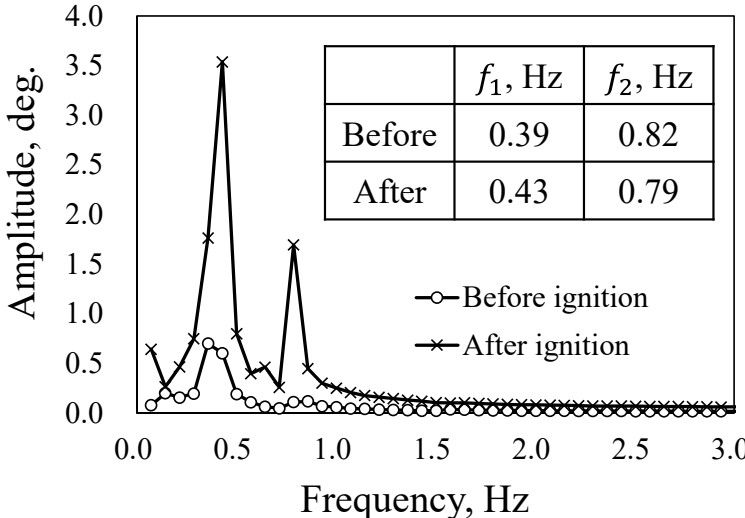

**Figure 5.** Spectrum of elevation angle before and after ignition.

To present a quantitative explanation of this behavior, the suspended launcher is represented as a double pendulum model, as shown in Figure 6. A rigid rod was suspended using a weightless wire with a rigid support. The mass, CG, and moment of inertia of the rod were the same as those of the launcher assembly. The wire connecting the launcher and hook has length $l_1$. The hook moves slightly in the horizontal direction; however, the hook is assumed to be a rigid support because of its significantly higher mass (60 kg) compared to the launcher assembly. This was further confirmed by the fact that the hook was almost stationary for several seconds after ignition (see Supplemental Video S2).

The eigenfrequencies of this model are obtained by solving Euler–Lagrange equations for the attitude angles.

$$\frac{d}{dt}\frac{\partial L}{\partial \dot{\theta}_i} - \frac{\partial L}{\partial \theta_i} = 0 \quad (i = 1, 2) \tag{4}$$

The Lagrangian is defined as $L = T - V$. The total kinetic energy comprises two terms of the translational motion of CG and the rotational motion around CG.

$$T = \frac{m}{2}\left(\dot{x}^2 + \dot{y}^2\right) + \frac{I}{2}\dot{\theta}_2{}^2$$
$$= \frac{m}{2}\left\{\left(l_1\dot{\theta}_1\sin\theta_1 + l_2\dot{\theta}_2\sin\theta_2\right)^2 + \left(l_1\dot{\theta}_1\cos\theta_1 + l_2\dot{\theta}_2\cos\theta_2\right)^2\right\} + \frac{I}{2}\dot{\theta}_2{}^2 \tag{5}$$
$$\cong \frac{m}{2}\left(l_1{}^2\dot{\theta}_1{}^2 + l_2{}^2\dot{\theta}_2{}^2 + 2l_1l_2\dot{\theta}_1\dot{\theta}_2\right) + \frac{I}{2}\dot{\theta}_2{}^2$$

Here the following approximations for small amplitudes of angles were used.

$$\sin\theta \cong \theta; \quad \cos\theta \cong 1 - \frac{1}{2}\theta^2 \tag{6}$$

The potential energy of CG is expressed as follows.

$$V = -mg(l_1\cos\theta_1 + l_2\cos\theta_2)$$
$$\cong \frac{mg}{2}\left(l_1{}^2\theta_1{}^2 + l_2{}^2\theta_2{}^2\right) \tag{7}$$

Here the approximations of Equation (6) were used again, and a constant term was eliminated because the origin of potential energy is arbitral. Thus, Lagrangian is expressed as a function of angles and angular velocities.

$$L = \frac{m}{2}\left(l_1{}^2\dot{\theta}_1{}^2 + l_2{}^2\dot{\theta}_2{}^2 + 2l_1l_2\dot{\theta}_1\dot{\theta}_2\right) + \frac{I}{2}\dot{\theta}_2{}^2 - \frac{mg}{2}\left(l_1{}^2\theta_1{}^2 + l_2{}^2\theta_2{}^2\right) \tag{8}$$

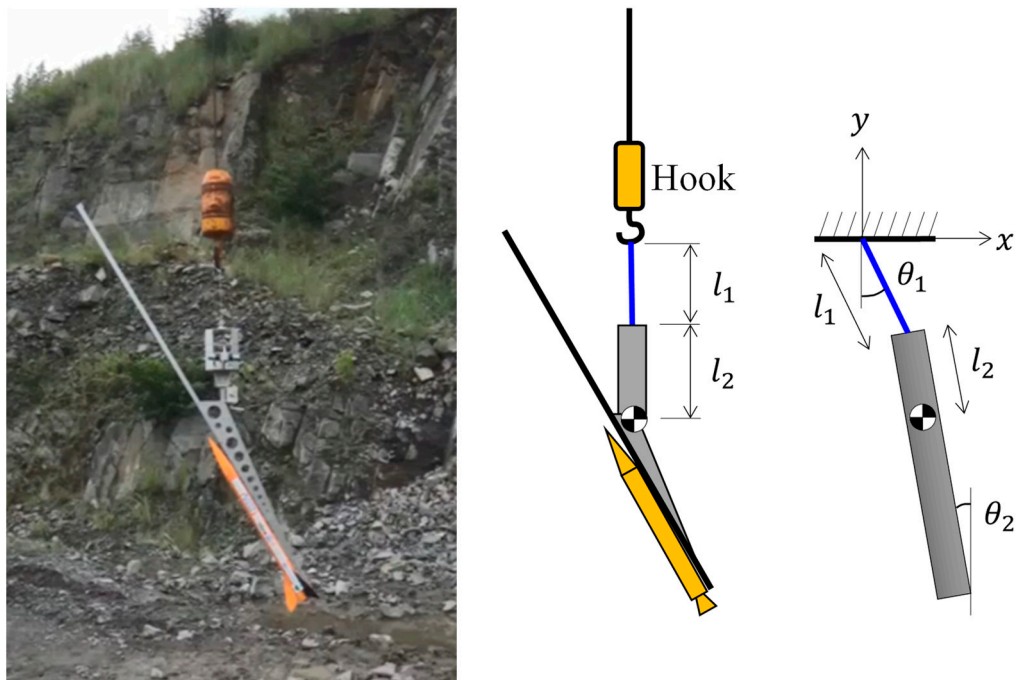

**Figure 6.** Double pendulum model of the suspended launcher. When the launcher is settled at the equilibrium state, $\theta_1 = \theta_2 = 0$.

Substituting Equation (8) to Equation (4), we obtain two motion equations.

$$l_1\ddot{\theta}_1 + l_2\ddot{\theta}_2 + g\theta_1 = 0 \tag{9}$$

$$l_1l_2\ddot{\theta}_1 + \left(l_2{}^2 + I/m\right)\ddot{\theta}_2 + l_1g\theta_1 = 0 \tag{10}$$

Now periodic solutions for the angles are assumed.

$$\theta_1 = \Theta_1 e^{i\omega t}, \qquad \theta_2 = \Theta_2 e^{i\omega t} \tag{11}$$

Substituting Equation (11) to Equations (9) and (10), we have a matrix equation.

$$\begin{pmatrix} -l_1\omega^2 + g & -l_2\omega^2 \\ -l_1l_2\omega^2 & -(l_2{}^2 + I/m) + l_1g \end{pmatrix} \begin{pmatrix} \Theta_1 \\ \Theta_2 \end{pmatrix} = \mathbf{0} \tag{12}$$

There is a non-trivial solution if and only if the determinant of the matrix is zero.

$$\frac{I}{m}l_1\omega^4 - \left(l_1l_2 + l_2{}^2 + \frac{I}{m}\right)g\omega^2 + l_2g^2 = 0 \tag{13}$$

There are two solutions for $\omega^2$. We take only positive roots for $\omega$.

$$\omega = \sqrt{\frac{mg}{2Il_1}\left((l_1 + l_2)l_2 + \frac{I}{m} \pm \sqrt{\left\{(l_1 + l_2)l_2 + \frac{I}{m}\right\}^2 - 4l_1l_2\frac{I}{m}}\right)} \tag{14}$$

The solutions $\omega_1$, $\omega_2$ are the 1st and 2nd mode frequencies ($\omega_1 < \omega_2$). The mode shape parameter is obtained by substituting Equation (14) to Equation (12).

$$\frac{\Theta_1}{\Theta_2} = \frac{l_2\omega^2}{g - l_1\omega^2} \tag{15}$$

The eigenfrequencies and mode shape parameter are calculated by using the actual values of structural properties as listed in Table 1. The mode shape parameters are positive for 1st mode and negative for 2nd mode. Hence, the mode shapes are as depicted in Figure 7. Theoretical 1st mode frequencies agreed with the experimental values presented in Figure 5, whereas those of 2nd mode frequencies are higher than the experimental values with 40% errors. However, they exhibit the same tendency that the 1st mode frequency increases and the 2nd mode frequency decreases after ignition. The eigenfrequencies were better matched with the experimental values by changing the parameters $l_1$ as presented in Table 1. The adjusted $l_1$ are larger than the actual values; this is because this model neglects the lateral motion of the crane hook suspended by the long wire. Actually, the crane hook moves slightly in the horizontal direction. The presence of a large mass above the launcher has a constraining effect on the pendulum motion.

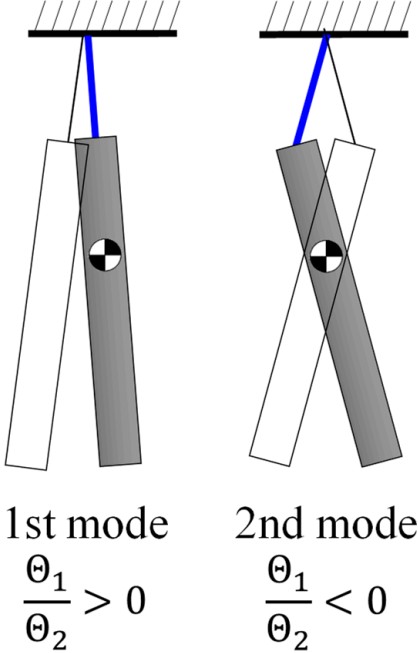

**Figure 7.** Double pendulum mode shapes.

**Table 1.** Structural properties and mode parameters.

| | | | Before Ignition | | After Ignition | |
|---|---|---|---|---|---|---|
| Mass (launcher + rocket) | $m$ | kg | 14.0 | | 12.8 | |
| Moment of inertia | $I$ | kg m$^2$ | 4.67 | | 2.96 | |
| | | | Actual | Adjusted | Actual | Adjusted |
| Length of the wire | $l_1$ | m | 0.357 | 0.950 | 0.357 | 0.950 |
| Distance of the CG | $l_2$ | m | 0.483 | $\leftarrow$ | 0.378 | $\leftarrow$ |
| 1st mode frequency | $\omega_1$ | Hz | 0.43 | 0.38 | 0.46 | 0.40 |
| Mode shape | $\Theta_1/\Theta_2$ | - | 0.49 | 0.60 | 0.47 | 0.61 |
| 2nd mode frequency | $\omega_2$ | Hz | 1.16 | 0.81 | 1.15 | 0.82 |
| Mode shape | $\Theta_1/\Theta_2$ | - | $-2.78$ | $-0.84$ | $-2.25$ | $-0.65$ |

The pendulum model can be improved without changing the parameters $l_1$ by adding another degree of freedom, i.e., crane hook and wire; however, a double pendulum model is used below because it is sufficient to represent the launcher's behavior in the experiment that only two frequency components are observed.

### 4.2. Elevation Angle Dynamics

The dynamics of the elevation angle were studied by considering a moving rocket with the pendulum model obtained above. First, the rocket was located at the bottom end of the launcher. After ignition, the CG and inertia moment change as the rocket moves on the rail. Considering the motion of the CG and the contact forces between the rocket and launcher, the equations of motion were solved.

As shown in Figure 8, a rail is fixed on a rigid rod with a target elevation angle $\varepsilon$. Fluctuation angle $\theta_2$ is defined as $\theta_2 = 0$ at the equilibrium state with the rocket on the launcher. When the rod is tilted with $\theta_2$, the actual elevation angle is $\varepsilon + \theta_2$ from the horizontal. The inertia properties of the rail are included in the rigid rod. Gravitational forces are considered for each CG of the launcher and rocket. A thrust force was applied at the tail of the rocket with a misalignment angle $\delta$. While the rocket with mass $m_r$ and inertia moment $I_r$ moves on the rail, two contact forces are considered between the rail guides and the rail, a tangential friction force $F$ and a normal force $T \sin \delta$. The rocket has front and rear rail guides. Because CG of the rocket is located nearby the front rail guide, the friction force was assumed to act only on this guide. The rocket is assumed to move on the rail at a uniform acceleration $a_r$ as follows:

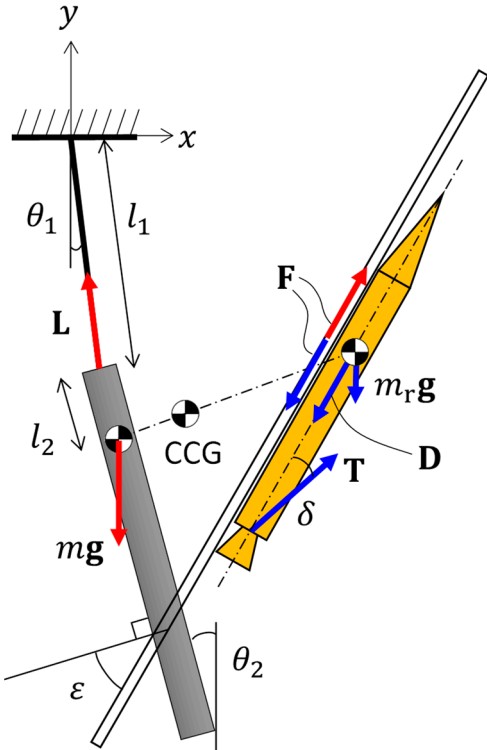

**Figure 8.** Double pendulum model of the suspended launcher with moving rocket. Red and blue arrows are acting on the launcher and rocket, respectively. The misalignment angle $\delta$ is highly exaggerated.

$$m_r a_r = T \cos \delta - F - m_r g \sin(\varepsilon + \theta_2) \tag{16}$$

The vertical motion of the CG of the launcher was neglected. The horizontal motion follows the equation of motion:

$$m\ddot{x} = F \cos(\varepsilon + \theta_2) + T \sin \delta \sin(\varepsilon + \theta_2) + L \sin \theta_1 \tag{17}$$

The last term is the contribution of the tension force $L$ of the wire, expressed as:

$$L = mg - F \sin(\varepsilon + \theta_2) + T \sin \delta \cos(\varepsilon + \theta_2) \tag{18}$$

The friction force is calculated if a friction coefficient $\mu$ was given:

$$F = \mu\{m_r g \cos(\varepsilon + \theta_2) + T \sin \delta\} \tag{19}$$

The launcher and rocket rotate around their combined center of gravity (CCG), which varies as the rocket moves on the rail. The combined moment of inertia $I_c$ around the CCG also varies over time. The external forces are tension $L$, gravitational forces acting on the launcher and rocket, and thrust **T**, which has a misalignment $\delta$. The friction forces are internal forces that cancel out in the rotational motion of the combined body. Additionally, D'Alembert's inertial force **D** is considered at the CG of the rocket with the magnitude expressed by Equation (16). The aerodynamic drag force was neglected because the velocity was low during rocket sliding on the rail. The rotational equation of motion is expressed with the external torque terms generated by these forces as follows:

$$I_c\ddot{\theta}_2 + Ll_2 \sin(\theta_1 + \theta_2) = mg(x - x_c) + m_r g(x_r - x_c) + (\mathbf{r}_T \times \mathbf{T} + \mathbf{r}_R \times \mathbf{D})_z \tag{20}$$

where $x$, $x_r$, and $x_c$ are the horizontal locations of the CG of the launcher, rocket, and their combination, respectively. $\mathbf{r}_F$, $\mathbf{r}_T$, and $\mathbf{r}_R$ are the relative location vectors from CCG, of the front rail guide, engine nozzle, and CG of the rocket, respectively. All of these location parameters vary with time. $\theta_1$ in Equation (20) is obtained from the following geometrical relation:

$$x = l_1 \sin \theta_1 + l_2 \sin \theta_2 \tag{21}$$

The horizontal location of the CG of the launcher $x$ is obtained by solving Equation (17).

The friction coefficient $\mu$ was obtained using the launcher and rocket used in the launch experiments. The launcher was prepared with the rocket on it in the same manner as in the launch experiments. The sliding surface of the rail was cleaned with a solvent before the silicone lubricant was sprayed on it. The launcher rail was tilted down to a negative elevation angle $-\theta$, until the rocket started sliding on the rail due to gravity. Then, the static friction coefficient was obtained as $\mu = \tan \theta$. The kinetic friction coefficient was obtained using the same equation, where $\theta$ is the smallest value at which the sliding rocket continues to slide without stopping. As a result, we obtained static $\mu = 0.18$ and kinematic $\mu = 0.12$. The latter value was used in Equation (19) because the rocket is sliding on the rail.

The thrust force was assumed to have a constant value of 60 N, which was obtained from the ascending motion of the rocket recorded in supplemental Video S1. The unknown thrust misalignment angle was varied to investigate its effect on the launcher dynamics. Figure 9 shows the calculated fluctuation angle of the launcher $\theta_2$ and its rate $\dot{\theta}_2$ compared with the experimental results, where the misalignment angle $\delta = 1.5°$ was selected for the best agreement. They are almost identical from the ignition time X to X + 1.3 s, that is, 1.0 s after the rocket has left the launcher tip. Furthermore, the following double pendulum motions also have the same frequency and phase, demonstrating the accuracy of the proposed model. Although the possible variation of the misalignment angle $\delta$ is unknown, the assumed value $\delta = 1.5°$ is reasonable.

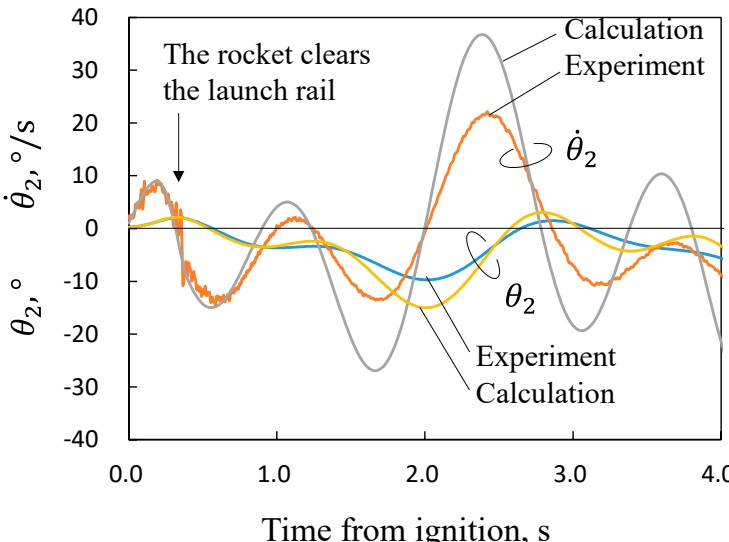

**Figure 9.** Double pendulum model of the suspended launcher.

### 4.3. Parameter Study

This model allows us to study the parameters that have a significant effect on the actual elevation angle behavior at launch. Among others, thrust misalignment, length of the launcher rail, and friction coefficient are considered here.

The thrust misalignment angle was assumed to be downward on the vertical plane, whereas the actual direction was arbitrary around the vehicle axis. A misalignment force in a horizontal plane causes a fluctuation in the azimuth angle, by which the trajectory and impact point turn around a vertical axis. A fluctuation in the elevation angle significantly changes the trajectory, apogee altitude, and downrange of the impact point. Figure 10 shows the fluctuation of the elevation angle for different angles of thrust misalignment, $\delta = \pm 1.5°$. The differences when the rocket leaves the rail are $\Delta \theta_2 = 2.8°$ and $\Delta \dot{\theta}_2 = 7.1\,°/s$. These variations significantly affect the launch trajectory. For example, a 5 $°$ difference in elevation angle results in a 26 km difference in the apogee altitude and a 100 km difference in the downrange of the impact points in the case of a single-stage suborbital mission with an apogee altitude of 100 km [17]. Although the effects of elevation angle rate $\dot{\theta}_2$ have not been investigated, they are not negligible.

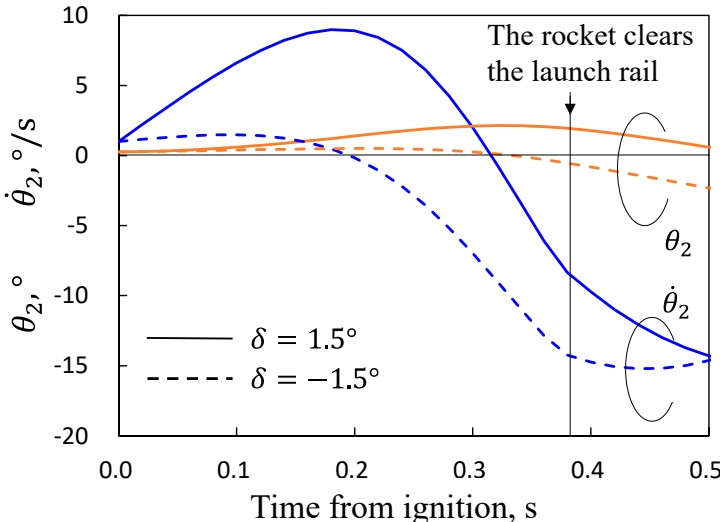

**Figure 10.** Elevation variations due to thrust misalignment.

These variations depend on the time when the rocket leaves the rail, which can be delayed by extending the launcher rail. However, it also has a nose-down effect because the downward torque due to the gravitational force of the rocket increases with the distance between the rocket and CCG. The relation between the rail length and the conditions when the rocket leaves the rail is shown in Figure 11. The launcher rail has to be long enough to obtain a sufficient velocity when the rocket leaves the rail, which is necessary for the aerodynamic stability of the rocket; however, an excessively long rail results in a large nose-down fluctuation, as shown in this figure. There is another tradeoff between the fluctuations of $\theta_2$ and $\dot{\theta}_2$: for the rail length with $\theta_2 = 0$, $\dot{\theta}_2$ has a large fluctuation, and vice versa. The optimum length that minimizes the trajectory error should be obtained through trajectory analyses with various combinations.

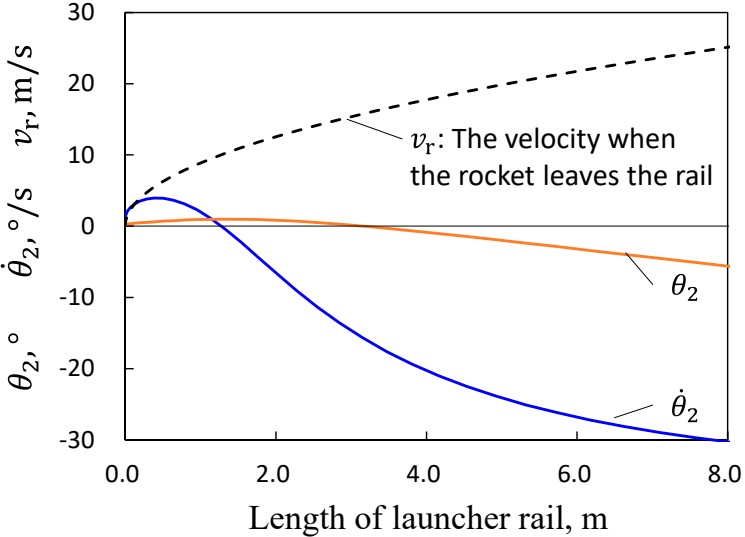

**Figure 11.** The conditions when the rocket leaves the rail and length of launcher rail.

The friction coefficient was obtained in the experiments mentioned above. However, it depends on the tribological conditions between the rail and rail guides on the rocket, such as surface roughness, cleanliness, and lubricant conditions. They could have a large variation depending on the preparation of the launcher and the meteorological conditions. When the silicone lubricant was not applied, the kinetic friction coefficient $\mu$ increased to 0.17. Figure 12 shows the elevation fluctuations with the thrust misalignment $\delta = 0$ comparing the friction coefficients. Considering a possible large value, $\mu = 0.50$ for unlubricated steels [19] was assumed. The increase in $\mu$ induces a larger nose-up torque during rocket sliding on the rail. The maximum elevation angle was delayed, and the peak value increased by $0.5°$. As a large value of $\mu$ causes these increasing variations in the elevation angle, it is necessary to minimize the variation of $\mu$ by cleaning and lubricating the rail. Although the friction torque can be eliminated by locating the CCG on the rail, the issue of thrust misalignment is more significant.

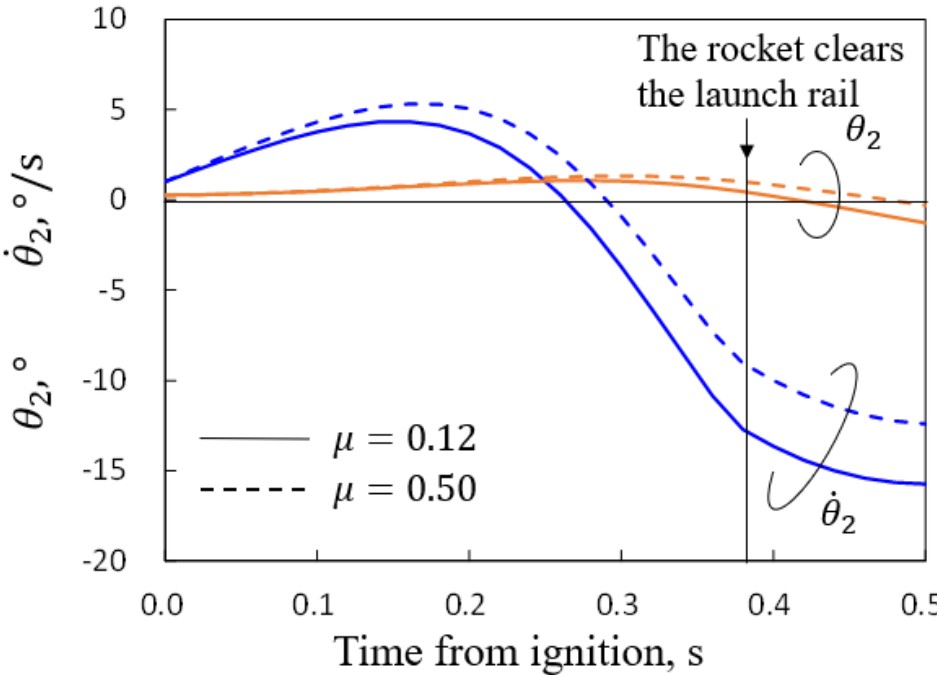

**Figure 12.** Elevation variations due to the friction coefficient without misalignment.

### 5. Conclusions

Small rockets were air-launched to demonstrate a suspended rail launcher equipped with a CMG, for application to a rockoon launcher. The attitude control of the azimuthal angle and dynamics in the elevation angle of the launcher were studied. The azimuthal control of the launcher was possible using the CMG. The following are concluded from the experiments:

- The azimuth angles of the impact points were within the range of $8°$ with respect to the target azimuth angle.
- The fluctuation of the elevation angle was observed due to the thrust misalignment and friction force between the rocket and rail. The launcher elevation angle should be determined considering these effects for accurate trajectory predictions.

The fluctuation behavior was explained using a double-pendulum model with a rigid rod. If a large mass exists between a launcher and balloon, such as an equipment box, then it has a constraining effect on the motion of the launcher, increasing the frequency of the pendulum motion. The error in the 2nd mode frequencies can be corrected by changing the wire length of the model without increasing the degrees of freedom. The following are concluded from the discussions:

- Rail lubrication is necessary to minimize variation in the friction coefficient.
- Thrust misalignment is the most significant cause of elevation fluctuation and also causes azimuthal error.
- There is a tradeoff between the leaving velocity and variations in the elevation angle and its rate when the rocket leaves the rail. The optimum length should be selected to minimize the trajectory error.

**Supplementary Materials:** The following are available online at https://www.mdpi.com/article/10.3390/aerospace8100289/s1, Video S1: Attitude controller unit test, Video S2: Air-launch experiment.

**Author Contributions:** Conceptualization, Y.W. and Y.F.; methodology, T.S and D.O; Investigation, N.K. and A.B.; formal analysis, T.S.; data curation, T.S. and D.O.; writing—original draft preparation, T.S.; writing—review and editing, Y.W. and A.B.; supervision, Y.W. and T.M.; project administration, Y.F. and T.M.; funding acquisition, Y.W., Y.F. All authors have read and agreed to the published version of the manuscript.

**Funding:** This research was funded by Yamaguchi Industrial Promotion Foundation.

**Acknowledgments:** We would like to thank the students at the Space Transportation Engineering Laboratory at Chiba Institute of Technology, who supported the preparation and operation of the air-launch experiments at Yamaguchi.

**Conflicts of Interest:** The authors declare no conflict of interest.

## Nomenclature

| | |
|---|---|
| $\omega$ | angular velocity of gyroscope wheels |
| $\theta$ | tilted angle of gyroscope wheel axis |
| $I_d$ | moment of inertia of the gyroscope rotor assembly |
| $I_L$ | moment of inertia of the launcher |
| $L$ | vertical component of the change in angular momentum of each gyroscope wheel |
| $\Omega$ | increase in the rotation speed of gyroscope |
| $M$ | torque generated by CMG |
| $l$ | length of crane wire above the hook |
| $l_1$ | length of wire under the hook |
| $l_2$ | distance between the top and the center of gravity of the rod |
| $f_1, f_2$ | eigenfrequencies of the 1st and 2nd modes |
| $\omega_1, \omega_2$ | angular eigenfrequencies of the 1st and 2nd modes |
| $\theta_1, \theta_2$ | fluctuation angles from the equilibrium attitude |
| $m$ | mass of rod |
| $I$ | moment of inertia of rod |
| $\varepsilon$ | target elevation angle |
| $\delta$ | thrust misalignment angle |
| $m_r$ | mass of rocket |
| $I_r$ | moment of inertia of rocket |
| $\mathbf{F}, F$ | friction force vector, and its magnitude |
| $\mathbf{T}, T$ | thrust force vector, and its magnitude |
| $\mathbf{D}$ | D'Alembert's inertial force vector |
| $a_r$ | acceleration of rocket |
| $\mu$ | friction coefficient |
| $I_c$ | combined moment of inertia |
| $x, y$ | horizontal and vertical locations of center of gravity (CG) |
| $\mathbf{r}_T$ | location vectors of the engine nozzle from combined center of gravity (CCG). |
| $\mathbf{r}_R$ | location vectors CG of the rocket from CCG. |

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
