# Peer review of "Air-Launch Experiment Using Suspended Rail Launcher for Rockoon"

_aerospace, doi:10.3390/aerospace8100289_

Round 1

Reviewer 1 Report

I liked this paper and the videos a lot, but here are some suggestions:

Errors to fix:

  • Add a space before “[9]”. (Line 55)
  • Change “leaving rocket” to “rocket leaving”. (Line 82)
  • Say “A genuine balloon experiment”. (Line 112)
  • Change “wheel” to “wheels” (Line 144)
  • Lines 178 through 192 should be erased.
  • Sometimes the degree symbol is next to the number, like in Line 237, but often there is a space, like in Lines 225, 241, etc. Be consistent. I recommend no space.
  • Change “the detail” to “a detailed” (Line 260) and pluralize “parameter” (Line 261).
  • Change “double pendulum model is used blow because” (Line 316 and 317) to “a double pendulum model is used below because”.
  • Don’t indent Line 348.
  • Say “degrees of freedom”. (Line 435)

Recommended changes/additions:

  • Is a “rockoon” a “method” of air-launching rockets (Line 10 and Line 41) or is it a physical device (including a launcher, a balloon, and (at first) a rocket) (as suggested by Line 79)? I think it is a device, so I suggest rewriting both statements that call it a “method.” It is certainly not a verb! (Line 45) Perhaps start that sentence with “The launch of a rockoon was first performed…”
  • Stick to past tense in the second paragraph. For example, change “must” to “had to” (Line 48) and “is” to “was” (Line 49).
  • Say “using a rocket launcher suspended from a balloon” rather than just “using a balloon” (Line 10 and Line 41).
  • In Figure 1 consider drawing the cable (and the hook?) above the CMG in the central drawing.
  • The term “slipper” is non-standard (used multiple times). Call it a “rail guide” or “rail button”.
  • The phrase “launcher clear” is non-standard (used multiple times, including in figure labels). Though it is cumbersome, the phrase “when the rocket clears the launch rail” is more appropriate. Similarly, “the leaving velocity” (Line 438) and “clear velocity” (Line 21) could both be changed to “the velocity when the rocket leaves the rail”.
  • Maybe change “via a crane car” to “using a crane”. (Line 206 and 207)
  • Consider changing “corrected” (Table 1 and Line 311) to “adjusted”.
  • Color would be useful in Figure 10 and Figure 11, just as it was helpful in Figure 9.

Some general comments:

  • The videos are neat but the link to them (Line 441) is broken.
  • Perhaps add more discussion about the effectiveness of the CMG in the discussion of Figure 4. I was surprised the Azimuth varied so much, even during the “controlled” times. The first video made it look like that angle could be held very steady.
  • Maybe add a notation on Figure 4 of when the rocket cleared the launch rail (at less than 1 sec).
  • There is a discrepancy between Figure 4 and the text below it as to what “X” means. If “X” is “time from ignition (in seconds)” then the text should say “X = -45” and “X = 0”, not “X-44 s” nor just “X”.
  • In Line 266 either put “s” everywhere or nowhere, consistent with the usage above.
  • I am not familiar enough with the math to check all the equations.
  • My biggest issue is with Figure 8. Why is the rail no longer aligned with the bar in the model (as was the case in Figure 6)? Are the forces meant to be acting (a) on the launcher or (b) on the rocket or (c) both? (Note: It isn’t a good idea to have free-body diagrams showing forces acting on more than one object.) The force F, for example, points in the wrong direction to be a frictional force acting on the rocket. If the forces are all supposed to be acting on the launcher then F is in the correct direction but T (thrust) should not be included because thrust acts on the rocket, not on the launcher. Also, perhaps explicitly state that δ is “highly exaggerated” in the figure.
  • I’m not familiar with the concept of “D’Alembert’s inertial force” but I hope the absence of force D in equations 16 through 19, but its inclusion in equation 20, is correct.
  • Claiming that the time derivative of the angle “is almost zero at …” (Line 416) is a bit of a stretch. On the graph it looks pretty far from zero for both frictional cases at that time.

Nomenclature list:

  • Add the word “gyroscope” to Lines 455, 456, 457, 459, 460
  • Change “over” to “above” in Line 462
  • Reconsider the explanatory text in Lines 465 and 466. Should they really be identical?
  • Spell out “CCG” in Line 481

Reviewer 2 Report

The paper is interesting also supported by data but it needs some minor improvements. 

The Discussion shall be reviewed in particular the elevation angle dynamic paragraph. The parameters are not clearly presented. 

The conclusion are not clear to the reader. 

I recommend a general review of the introduction section and an extensive english review . 

Reviewer 3 Report

Dear Authors,

I find this manuscript interesting due to limited literature on this topic. I have reviewed this work earlier for a different journal (before some modifications), but I find that it has value despite that it has not been published. I suggest a minor revision of the present manuscript. Please find my comments and questions in the following points.

I have the following comments:

  • The literature review is satisfactory - it mentions examples of rockoons and their launch system types
  • Some more background on the study and your new rockoon system would be interesting. What is the role of the different entities involved (CIT and 3 companies are mentioned within the authors’ affiliations)?
  • The azimuth was maintained within +/- 8 deg (Fig. 4) – is this a value that can be assumed easy to repeat? How can it be improved? What was the situation during the other two flights? What would the +/- 8 deg accuracy of azimuth mean for a full-scale rockoon mission concerning landing points, safety?
  • A paragraph on potential disturbances during a full scale mission would be beneficial in the first part of the manuscript. What effect would wind gusts have?
  • Line 36 – a reference showing that the idea came from the High Virgo missile would be appreciated
  • Line 39 – “liquid-propellant orbital rocket” instead of “liquid-fueled orbital rocket” is suggested
  • In lines 89-97 the temperature of N2O storage is mentioned. What range of altitudes and temperatures are considered? Will they always (in each mission) be in a relatively narrow range, to ensure similar N2O density and feeding pressures (if self-pressurization is used)? Side remark: heat conductivity may be low due to similar temperatures, but ideal equilibrium is difficult to achieve.
  • Line 90: a few sentences on the size and type of potential launch vehicle would be very valuable (highly innovative approach)
  • Please mention Fig. 8 in the text before it occurs in the file
  • Lines 178-192 are not part of the manuscript, they seem to be taken from the template
  • Please mark your changes to the manuscript if submitting the revised file

Yours sincerely,
